# The Correlation of Tooth Sizes and Jaw Dimensions with Biological Sex and Stature in a Contemporary Central European Population

**DOI:** 10.3390/biology13080569

**Published:** 2024-07-28

**Authors:** Kurt W. Alt, Nils Honrath, Maximilian Weykamp, Peter Grönebaum, Nicole Nicklisch, Werner Vach

**Affiliations:** 1Center of Natural and Cultural Human History, Faculty of Medicine and Dentistry, Danube Private University, Förthofstrasse 2, 3500 Krems-Stein, Austria; nils@zahnarzt-honrath.de (N.H.); maximilian.weykamp@hotmail.com (M.W.); peter.groenebaum@dp-uni.ac.at (P.G.); nicole.nicklisch@dp-uni.ac.at (N.N.); 2Institute of Prehistory and Archaeological Science, Department of Environmental Sciences, University of Basel, Spalenring 145, 4055 Basel, Switzerland; werner.vach@unibas.ch

**Keywords:** tooth size, jaw dimensions, stature, biological sex, correlation analysis

## Abstract

**Simple Summary:**

Over the course of the past 3 million years, changes in human habitats, nutrition and behaviour have led to continuous processes of shape modification and size reduction in our jaws and teeth. These have resulted in the specific masticatory system observed today and the widespread need for orthodontic treatment. The aim of this study was to identify correlations between tooth sizes, jaw dimensions, biological sex and stature in a central European sample. We observed differences between women and men with regard to their tooth and jaw dimensions. In general, men’s teeth, especially the canines, are larger. There are also differences in tooth sizes and jaw dimensions in relation to stature, with the canines again standing out. Our results help us to better understand the evolutionary trends affecting our teeth. They also indicate that, while the basic sexual dimorphisms of our primate ancestors have been retained, modern dietary habits will further intensify the reduction in the masticatory apparatus. Furthermore, the results provide sex-specific metric data of forensic significance, as well as clues as to why more and more people require orthodontic treatment, while suggesting possible causes for diachronic differences in tooth and jaw dimensions between populations.

**Abstract:**

Dental anthropology provides a deep insight into biological, ecological and cultural aspects associated with human individuality, behaviour and living conditions and the environment. Our study uses a correlation analysis to test the metric relationships between tooth sizes and jaw dimensions and juxtaposes them with biological sex and stature. A sample of *n* = 100 dental casts was used to record metric dental data including the mesio-distal and bucco-lingual tooth crown diameters and nine upper and lower jaw dimensions. All crown diameters were highly correlated with both stature and biological sex, with the canines exhibiting the highest correlation. The majority of jaw dimensions exhibited similar correlations. Our results suggest that the differences between the sexes in most crown diameters and some jaw dimensions may be related to the stature of the individuals measured. Two groups of closely correlating features emerged among the jaw dimensions, differing in their degree of correlation with crown diameters and with sex. The results and insights obtained are highly relevant for evolutionary biology, dentistry, craniofacial research, bioarchaeology and forensic odontology.

## 1. Introduction

The scientific research into the shapes and sizes of our teeth has a long history. As part of his twin studies, G. Korkhaus identified a higher variability in tooth sizes compared to their shapes [1]. It is generally accepted that both the shapes and sizes of dental crowns and roots and jaw dimensions are genetically predetermined and exhibit differences, both between the sexes and between populations [2,3,4]. Tooth sizes and jaw dimensions are controlled by hereditary factors. They appear to be ontogenetically connected because the occlusal relationships between the maxilla and the mandible depend on tooth sizes and jaw dimensions being properly aligned with each other [5]. Insights into the ranges of tooth sizes and jaw dimensions are important for orthodontic treatments, evolutionary research, bioarchaeology and dental forensics [6,7,8]. Disorders that manifest in dentition during ontogenesis occur as a result of genetic and epigenetic impacts on the phenotype. Anomalies such as hyper- and hypodontia, mismatched tooth and jaw dimensions, tooth displacements and jaw malpositions all have an impact on the harmony of the face and, as a consequence, on a person’s appearance and individuality [3,9].

Humans have continued to evolve since they first appeared three million years ago, though selection pressure on *H. sapiens* has decreased in the past 12,000 years, i.e., since the beginning of the Holocene [10]. Over the course of evolution, changes to habitats and functional behavioural adjustments have inevitably led to processes of adaptation in all species. As a result of the phylogenetic history of hominins, modern humans were also part of this process. Besides bipedalism, increased brain size and adaptations to the hands, the gradual regression of the jaw, which originally protruded in the shape of a snout, and the associated reduction in tooth size demonstrate the special evolutionary path of humans [11]. Once *H. erectus* left Africa approximately 2 million years ago and began to move into Eurasia, the most distinct feature they had was a reduction in tooth size [12]. This was due, in part, to new food preparation techniques and, probably, to the use of fire [13,14]. Thanks to innovations such as fermentation and the use of cooking vessels, food was refined even further. Cooked meat, tubers and vegetables generally put less pressure on the teeth and jaw bones [15]. Eventually, even people who had lost all their teeth were able to survive [16].

An interesting aspect in terms of tooth size is that, in the past, populations that were highly dependent on heating food for their survival all had smaller teeth. This is still reflected in a north–south gradient within Europe today, in that populations in the north are more dependent on fire due to the climate than those in the south [15]. From a genetic point of view, mutations and a growing population density over the past 45,000 years have resulted in a decrease in tooth sizes throughout central Europe [17]. The relationship between tooth sizes and jaw dimensions has also changed over the course of human phylogeny [5].

Taking into consideration the frequency of dental overcrowding in recent humans, the reduction in tooth sizes appears to have had a greater impact on the jaw than on the teeth [18]. This diverging masticatory force is explained by differences in size between different types of teeth and their unequal adaptation to modern living conditions. Premolars and molars are exposed to greater selective pressure, in this respect, than anterior teeth. The decrease in the size of anterior teeth, on the other hand, can be seen as an adaptation to the change in the available space [19]. Over the past 10,000 years, the reduction in tooth sizes has proceeded twice as fast as before [15]. The transition between an appropriative and a productive way of living during the Neolithic period and the beginning of a sedentary lifestyle played a crucial role in this, as has, more recently, industrialisation and its impact on our diet and eating habits [20].

Our study examined the association of crown diameters and jaw dimensions with both stature and biological sex in a contemporary central European population. Standardised data relating to the teeth and jaws were collected using anonymous dental casts. The findings have been interpreted against the background of human evolution and the lifestyle changes described above. However, such associations may be due, in part, to the sexual dimorphism affecting both tooth sizes and jaw dimensions or to stature, which again is known to show sexual dimorphism [21]. Hence, we must take all these factors into account. We will therefore begin by describing the association of tooth sizes and jaw dimensions with sex and with stature and illustrate the joint contribution of sex and stature to explaining the development of crown diameters and jaw dimensions. We will then investigate the associations between crown diameters and jaw dimensions which cannot be explained by stature or sex.

It is a well-known fact that there is a more or less pronounced sexual dimorphism in primates and that this includes the teeth [22]. Studies on human teeth show overlapping dimensions for both sexes, with significant differences in their mean values, where teeth are generally larger in males than in females [23,24,25,26,27]. In all studies, the canines exhibit the greatest sexual dimorphism [23]. Apart from these well-known sex-related differences, a correlation between tooth sizes and stature is also being discussed [28,29,30]. Again, it is mainly the upper canines that appear to provide the most distinct results in this regard [30]. In contrast, the third molars were excluded from this study. The prevalence of third molar agenesis is quite high globally, ranging from 9.7% in Canada to 41% in Korea [31]. Moreover, third molars vary greatly in shape and size [3].

## 2. Material and Methods

This study was based on data from 100 students aged between 20 and 35 at Danube Private University (DPU), Krems-Stein, Austria. The available information included the subject’s name, sex, year of birth and stature. For the analysis itself, the data were anonymised, and the original information was accessible only to the study director. In order to better assess and validate the sexual dimorphism in the test subjects, we were mindful of a balanced sex ratio (50 males and 50 females). Two of the study subjects were brothers but not twins. A total of 92 were of central European origin, the others were from Eritrea, Vietnam, Greece, Russia (*n* = 2) and Iran (*n* = 3). The study was conducted following the guidelines established by the Declaration of Helsinki. Ethical approval was granted by the Ethics Committee of the Danube Private University, Krems, Austria (DPU-EK/027/2023).

Alginate impressions of the test subjects’ jaws were taken under supervision at the DPU’s outpatient clinic and casts, in class II dental plaster, were prepared under the guidance of the university’s dental laboratory staff. The majority of the test subjects had a full set of teeth. Only a few individuals (*n* = 5) were missing premolars due to orthodontic procedures. A total of 10 subjects were missing at least one tooth, including third molars. One test subject was undergoing orthodontic treatment when the impression was taken and was wearing dental braces. Due to the limited age range, the dental charts of the study participants were readily comparable, and the incidences of dental treatment were low, which meant that the measurements carried out as part of the study were not significantly impacted by external influences such as occlusal or approximal changes in dental hard tissues due to abrasion/attrition or dental treatments (dental fillings, prosthetics). In addition to tooth and jaw dimensions, stature was the third metric variable recorded. The age distribution of the study participants ensured, on the one hand, that they had reached their full stature and, on the other, that a reduction in stature due to wear, as seen in subjects of advanced age, could be excluded.

The dental crown diameters and jaw dimensions were measured using international standards [32,33,34]. In order to identify potential correlations between various metric features such as the length, width and depth of the jaw, and between the jaw and tooth dimensions, eleven measurements were developed with the aim of defining the jaw’s horizontal, sagittal and vertical dimensions. Nine of the dimensions referred to the jaw, and two to the teeth (Table 1). The points of measurement were characterised by anatomically defined reference points or areas that served as starting and end points when measuring distances and lines. Some of the jaw dimensions were modified in such a way that they could be applied to the test subjects’ dental casts. Table 1 provides an overview of the measurements used. When taking the jaw measurements, the shortest distance between two measuring points was usually determined. The mesio-distal (MD) and bucco-lingual (BL) crown diameters of all teeth of the permanent dentition, except for the third molars, were measured. The MD and BL distances represented the maximum extension of each tooth in the direction measured [35].

The potential sources of error when carrying out such measurements are generally diverse and numerous. They range from technical problems with measuring devices to artificial and pathological changes in the teeth, methodological mistakes due to the subjectivity of the measuring process, transcription mistakes and clerical errors to mistakes made when taking the jaw impressions or when casting the plaster models. In order to reduce the impact of intra- and interobserver variability, all measurements were repeated twice (jaw dimensions) or three times (crown diameters). Two different observers were involved in the assessment. The measurements were taken using digital callipers with an accuracy of 0.01 mm. All values were averaged over repeated measurements. In addition, crown diameters were averaged over the left and right sides, as none of the jaw dimensions were side-specific. As a result, 37 variables were included in the analysis: the MD and BL diameters of seven teeth (teeth 1 to 7 = central incisors to second molars), both in the upper jaw (UJ) and in the lower jaw (LJ), and nine jaw dimensions (Table 1). The stature and sex of each participant were also included in the study.

The first step was to provide information about the potential measurement errors, followed by determining the correlations between each variable and sex and stature. The next step was to show the relative role played by stature and biological sex in the intra-population variation. The main part of the analysis explored the correlations between the nine jaw dimensions and the crown diameters, partialling out the association with stature and sex. Since the jaw dimensions potentially reflect latent variables representing the true morphological jaw genotype, we also attempted to determine these latent variables by means of a factor analysis and included corresponding factor scores when analysing their association with the crown diameters.

Information on the measurement error of each variable was provided by reporting the estimated standard deviation (standard error) of the raw measurements using the information provided by two, three and six repeated measurements, respectively. These estimates were based on a random effects model using the individual and—in the case of the crown diameters—the side of the jaw as random effects, allowing for varying differences between the two sides from one individual to the next. In addition, the measurement error of the variables analysed was depicted by the standard error (SE), coefficient of variation (CV) and intra-class correlation (ICC). The latter equals the correlation expected when repeating the entire measurement procedure. The population distribution within each sex was described by means and standard deviations. The sexual dimorphism was assessed by the ratio between sex-specific mean values. The association between the variables and stature was assessed using the Pearson correlation coefficient [36].

A regression model relating each variable to stature and/or sex was used to determine how much of the population’s variation could be explained by these two variables. We have reported the adjusted R^2^ values of stature only, sex only, or both, and the *p*-value of the null hypothesis of no effect of sex in a model with both covariates. The link between jaw dimensions and crown diameters was depicted by correlation coefficients partialling out the effects of stature and sex. This was achieved by basing the correlation on the residuals from regressing each variable on stature and sex, i.e., by considering how much a variable is actually above or below the expected value based on an individual’s stature and sex. A factor analysis with a varimax rotation was applied to the residuals to identify potential latent variables. The number of factors was determined as the minimum number explaining at least 90% of the variance. Factor scores were used to obtain approximate values for the latent variables.

## 3. Results and Discussion

### 3.1. Basic Study Population Characteristics

Stature ranged from 171 cm to 199 cm in males and from 158 cm to 183 cm in females. The average stature was 182 cm in males and 169 cm in females and was therefore larger than the population averages in Germany (180/166) and Austria (178/166), the countries from which the majority of students hailed [37]. In addition to hereditary influences, a high standard of nutrition and health care are among the selective factors for above-average body height.

### 3.2. Measurement Error

The standard error of single measurements of the crown diameters typically ranged between 0.1 and 0.2 mm. The standard error of the variables used in our analysis (averaging over three replicates and two sides) was usually around 0.1 mm (Table 2). The coefficients of variation were typically below 2% and ICCs were always above 0.9, indicating a high ability to differentiate between subjects with differences in their true crown diameters. The jaw dimensions had larger measurement errors than the teeth. However, given their larger size, similar coefficients of variation were obtained. Their ICC values were even more favourable.

### 3.3. Sex-Specific Distribution of Crown Diameters and Jaw Dimensions

Table 3 shows the distribution of all crown diameters and jaw dimensions within the study population by sex. We can observe the well-known pattern of the smallest diameters for the incisors and largest diameters for the molars. Studies on both prehistoric and recent populations have found that males generally have larger teeth than females [23,27,38,39,40], which our study confirmed. The ratios between the mean values in males and females varied between 1.01 and 1.08, with an average of 1.04. This 4% difference in the average tooth size matched the results of a meta-analysis carried out using data from modern central European populations that was published recently [27]. The largest sexual dimorphism was observed in the canines, which again corresponded with the existing literature [13,27,41,42,43]. Larger BL diameters of the tooth crowns in males compared to females in contemporary populations have been explained by higher percentages of dentine by weight [24]. The authors of that particular study suggested that this could be used in the future for sex determinations. Although this is an interesting idea, it has not been followed up on so far.

Research on the possible links between jaw dimensions and biological sex is less exhaustive [44,45]. From the point of view of primate evolution, metrically useful parameters such as jaw length (prognathism) and tooth width do exhibit sexual dimorphism but they are generally associated with weight rather than stature [46,47]. In our study, the jaw dimensions were all greater in males compared to females, with ratios between 1.02 and 1.07, except for *palatal height*, which had a ratio of 1.16. These results are similar to those found in the literature, although greater differences have also been reported [44].

### 3.4. Relation of Crown Diameters and Jaw Dimensions to Stature

All crown diameters showed a correlation with stature, typically at a magnitude of 0.40 (Table 4). The association between stature and crown diameter is very similar between the upper and lower jaws (average correlations of 0.39 and 0.38, respectively) and between BL and MD diameters (average correlations of 0.40 and 0.38, respectively). The association was most pronounced in the canines (average correlation 0.50) and least obvious in the second incisors and second molars (average correlations of 0.31 and 0.34, respectively).

Several studies have recorded a link between tooth dimensions and stature in early hominids [48], various primate species [49,50] and modern humans [30,51,52,53,54,55]. Most publications report a positive link between tooth size and stature, though some studies have identified a rather low correlation [53,56,57]. The latter can be explained by specific environmental conditions, such as in the Baka Pygmies, where their phenotype is adapted to life in the forest [53]. In principle, dentition should only be used as an addition to more robust indicators of stature [56].

To the best of our knowledge, the association between jaw dimensions and stature has never been investigated. Our study identified correlations of around 0.4, except for *anterior palatal height*, which showed no correlation. Two dimensions (*anterior mandibular width* and *external dental arch width*) showed correlations of above 0.5, exceeding all values observed for the crown diameters.

The correlations between stature and crown diameters or jaw dimensions appear to be less pronounced than those of other parameters that have been reported in the literature and are traditionally used for stature reconstruction, e. g. the long bones in the lower extremities. This is probably due to the fact that the ontogeny of long bones takes a relatively long time, and we can therefore expect their higher correlation with stature. As a consequence, researchers have previously concluded that dentition should only be used as a means of determining stature if it is combined with other, more robust indicators [58]. In general, combining several traits obviously increases the reliability of stature estimations, and no single parameter should be drawn upon when studying fragmentary remains [59]. Bioarchaeological genome-based stature reconstructions have highlighted the methodological potential of this type of study [60]. However, it should be noted that stature and health are based on a complex combination of biological, behavioural and environmental factors [21]. As forensic research is often forced to contend with severely fragmented skeletal remains or even just with skulls, jaw dimensions may be another useful set of parameters [61,62].

### 3.5. Variations in Crown Diameters and Jaw Dimensions by Stature and Sex

The fact that crown diameters and jaw dimensions correlated with both stature and biological sex in our study obviously raises the question of how a combination of factors impacts tooth and jaw dimensions. In view of the known link between sex and stature and the expected correlation between stature and the size of skeletal elements, we must question whether the sexual dimorphism in tooth and jaw dimensions is exclusively based on the sexual dimorphism in stature, or whether biological sex plays an additional, unrelated role. To the best of our knowledge, this question has never been addressed before.

In our study, stature explained between 5% and 23% of the variation in crown diameters, while 1% to 23% of the variance was explained by biological sex (Table 5). However, when considering the impact of stature and sex combined, hardly any increase was seen compared to the effect of stature alone, suggesting that stature was the main or even the sole mediating factor in the influence of sex on the crown diameters of most teeth. A statically significant effect of sex on its own was only observed in a few instances for some of the incisors and canines. In general, the canines were the teeth for which the intra-population variation could best be explained by stature and/or sex.

The picture was more heterogeneous with regard to the jaw dimensions. For some jaw dimensions, the link to height or sex was rather tenuous. However, more than 25% of the variation in the *anterior mandible width*, *external dental arch width* and *palatal height* was explained by stature and sex combined. Again, the additional impact of sex alone was generally quite small, but it was demonstrated for some of the jaw dimensions (*external dental arch width, anterior mandible width, palatal height*).

If the association of some crown diameters and jaw dimensions with sex can be explained by a joint association with stature, this raises the question of whether it is adequate to regard them as sex indicators. Of course, we can still justifiably speak of a sexual dimorphism here, as biological sex can be viewed as a contributory factor to the differences observed. However, it may be more correct to see them as stature indicators which also allow us to make inferences about biological sex.

Furthermore, when speculating about an independent impact of biological sex on the tooth sizes that cannot be explained by the link between sex and stature, it might be useful to take into consideration the early and relatively short period of dental crown formation. It predates by far the various phases of long-bone growth and, therefore, the moment at which full stature is reached and offers relatively little opportunity for environmental factors to have an impact. According to genome-wide association studies, stature is highly hereditary in humans [63] and is dependent on polygenetic and epigenetic factors [64,65]. The unlocking of genetic potential, however, is significantly influenced epigenetically by the living conditions prevalent during the development phase [66]. Individual or population-specific variations in living conditions and dietary habits could be partly responsible for the considerable ethnographic differences in the data [21]. Selection pressure, or adaptation to the environment, is one of the decisive factors with regard to the correlation between stature and tooth sizes. Evidence to support this was obtained, for example, by a study carried out on a population of Baka Pygmies [53], who are greatly affected by environmental conditions, where no link was observed between their tooth sizes and stature. While these considerations do not apply to jaw dimensions, we were able to demonstrate an independent impact of sex on three jaw dimensions in our study.

### 3.6. The Correlations between Jaw Dimensions

Appendix A shows the correlation matrix of the jaw dimensions after partialling out the effects of sex and stature. We can observe moderate to high correlations between some dimensions (e.g., between the *dental arch lengths* in the upper and lower jaws and the *internal palatal length*), whereas other pairs of dimensions seem to be more or less independent (e.g., *palatal end width* and *internal palatal length*). Every dimension exhibits at least a moderate correlation with some, though not all, other dimensions, suggesting an underlying latent factor structure.

A factor analysis showed that more than 90% of the variance could be explained by two latent factors. The loading matrix of these two factors is shown in Appendix A. *Palate end width, anterior palatal width* and *external dental arch width* load on the first factor. *Internal palatal length, anterior mandible width* and both *dental arch lengths* load on the second factor. *Palatal height* and *anterior palatal height* do not show a clear relation to any of these two factors, whereas the *anterior mandibular width* shows some association with both factors.

### 3.7. Relation between Tooth Diameters and Jaw Dimensions and the Interpretation of Latent Factors

A distinct association between jaw dimension and (almost all) tooth diameters could be identified for the *internal palatal length*, the *anterior mandibular width* and both *dental arch lengths* (Table 6). As these four dimensions were all loaded on the second latent factor, this factor also showed moderate to high associations with the tooth diameters. In contrast, the first latent factor showed very low correlations. Besides the jaw dimensions loading on the second latent factor, only the *external dental arch width* reached a moderate correlation with some of the tooth dimensions. The jaw dimensions *internal palatal length*, *anterior mandibular width* and *dental arch length* loading on the second factor are all related to the space available for the teeth. The fact that the second factor is distinctly correlated with the tooth diameters suggests that the jaw acts as a place-holder for the teeth. Obviously, a larger jaw offers more room for bigger teeth and if the jaw is not long enough this will lead to overcrowding. This is further supported by the observation that the average correlation of the second latent factor with the MD parameters (r = 0.56) was more pronounced than its average correlation with the BL dimensions (0.52); a relation which also holds true when the upper and lower jaws are studied separately and also applies to 10 out of 14 comparisons at the level of single tooth measurements. This probably shows that the MD diameters are limited by the jaw’s length. The BL crown diameters are not restricted in the same way, and they exhibit a lower, though distinct, correlation. The similarity of the correlations between the MD and BL diameters also suggests that the link is not functional (i.e., teeth are smaller because they have less room to develop) but that it is determined by genetics. Studies of primates have confirmed the high correlation between features of the facial bones and molar sizes [67]. Studies on mice have yielded similar results, although researchers there associated the molar configuration, the shape of the mandible and various skull dimensions with the impact of closely related quantitative loci [68].

The two latent factors also vary with regard to the degree of sexual dimorphism. The four jaw dimensions loading on the second factor show ratios between 1.02 and 1.05 (Table 3), i.e., a limited degree of sexual dimorphism compared to other jaw dimensions. In addition, the data failed to demonstrate an impact of sex on these dimensions beyond the impact of stature (Table 5). In contrast, the three variables loading onto the first factor showed ratios between 1.05 and 1.08, i.e., a higher degree of sexual dimorphism (Table 3). Moreover, for two of the dimensions, an additional impact of sex over and above the impact of stature could be demonstrated (Table 5). The jaw dimension *palatal height* exhibited the greatest sexual dimorphism (Table 3) (loading 0.235, Appendix A). This suggests that the first factor must be interpreted as perceived facial masculinity [69], which does, in fact, interact with stature [70]. The importance of *palatal height* was demonstrated by a Swedish study on the patients of an orthodontic clinic. Subjects with high palates were significantly taller than other Swedes of the same sex and age. Hereditary causes such as Marfan syndrome or thump-sucking habits, which are often associated with high palates, did not apply [71].

### 3.8. Evolutionary Oral Adaptation with Transdisciplinary Implications for Evolutionary Biology, Bioarchaeology, Dentistry and Forensic Odontology

Our study examined the correlations between tooth sizes and jaw dimensions and how they are linked to stature and biological sex. From an ontogenetic point of view, there are various reasons why tooth sizes are particularly suited to population genetic studies. Their features are easy to quantify and the fact that teeth are formed over a short period of time means that environmental factors have a limited impact.

From the point of view of evolutionary biology, it is important to note that the gradual reduction of the “snout” in the primates that evolved into the genus *Homo* involved a shortening of the jaw and a reduction in the number and size of teeth, the canines in particular, as Charles Darwin observed [72]. In the species *Homo sapiens,* and throughout the course of human evolution, this resulted in a dentition with distinctly shortened parabolic dental arches, relatively small incisors and canines and a differentiation of the posterior teeth [73]. Because tooth and jaw dimensions are inevitably interdependent, both anatomically and functionally, one would expect both traits to exhibit largely interdependent behaviours. However, the prevalence of dental overcrowding in recent populations appears to contradict this assumption [35].

Nevertheless, dental overcrowding does seem to be a consequence of the modern lifestyle, where the “locking” of the teeth into a position of maximal intercuspation appears to be linked to a lack of occlusal or approximal adjustment and adaptation processes in the dental hard tissues. Until well after the Middle Ages, dental crowns were reduced both in height and width by masticatory activity as individuals aged [74]. The almost complete lack of dental crown wear due to modern living conditions has its consequences. As far as it affects the points of contact between the teeth, the lack of wear results in an increase in space-related anterior crowding. This would mean that dental overcrowding is a functional phenomenon. Some studies, however, discuss hereditary components in the occurrence of overcrowding [75]. The heredity of various cephalometric variables including the length of the maxilla and mandible has clearly been documented in twin studies, while dental variables such as tooth sizes are generally more closely linked to environmental factors [76,77].

The original purpose of teeth, which was to crush and break up food, became increasingly less important thanks to the use of tools, the cooking of food (meat, plants, tubers) and other innovations such as the use of vessels, while the masticatory pressure on the teeth and jaw continually decreased [15]. Other far-reaching changes that have occurred since the beginning of the Holocene (Anthropocene), such as the transition to a productive economy (12,000 BC) or the Industrial Revolution (250 years ago), again brought about profound changes in the dietary habits of *H. sapiens* [78]. Ultimately, it was changes in living conditions as part of cultural evolution that had a selective effect and resulted in the trend towards smaller teeth [19,79,80].

To what extent our results have general validity in other circumstances is a difficult question to answer. There is no doubt that there has been a fundamental and distinct trend towards a reduction in tooth size in *Homo sapiens* since the Neolithic period, which can be attributed to a decrease in selection, though its onset was much earlier, around 100,000 years ago, in the Pleistocene [12,15,53,81]. The existence of a sexual dimorphism is not disputed [24]. Tooth size variation in contemporary populations shows large global differences [39,82]. The extent of this variation differs between major geographic regions and ethnic groups, whereby population affiliation and lifestyle (e.g., a Western diet) play a decisive role [82,83]. With regard to stature, a recent study has pointed to the large significance of socio-cultural factors in the development of sexual dimorphism [21].

### 3.9. Strengths and Limitations

Our study demonstrates that a systematic analysis of the association between tooth size, jaw dimensions, stature and biological sex can provide a series of insights: (1) tooth crown diameters and jaw dimensions show a pronounced sexual dimorphism; (2) crown diameters and jaw dimensions correlate with both stature and biological sex; (3) stature seems to be the main mediating factor in the influence of biological sex on dental metrics; (4) the sexual dimorphism of teeth remains a decisive factor in the differences observed. The results of the jaw dimensions suggest that they may be as useful, or perhaps even more useful, than crown diameters when it comes to sex and stature determinations. Further research will be required to clarify their value both in this and in other respects.

When considering sex and stature as predictors of tooth sizes and jaw dimensions, our results indicate that the link between tooth size and biological sex can often be explained by a shared link with stature, while there is little evidence for a substantial role played by sex on its own. This should be taken into account when interpreting the sexual dimorphism in tooth sizes and jaw dimensions. It might be advisable in future to regard many of these variables as stature indicators, which, based on the link between stature and sex, can also serve to determine the latter. It is an open question whether the same also applies to other sex indicators.

Limiting the study subjects to a socially defined group of people (e. g. medicine students) can, on the one hand, result in a lower variation within the study group, which would then suggest that the correlations in the overall population would be higher. On the other hand, it is also possible that additional factors would come into play when studying the overall population, which in turn would result in lower correlations. Similar studies using different recent population groups would therefore be very interesting but are difficult to carry out. In principle, the results presented by us and others [83,84,85] confirm the importance of the dental metric parameters analysed when it comes to answering specific questions concerning evolutionary biology, dentistry, dental anthropology and forensic odontology.

## 4. Conclusions

Accurate sex and stature determinations play a fundamental role in biological and medical applications and research. Our study has improved the approach to formal reconstructions and our ability to interpret individual estimates. When interpreting associations between tooth sizes, jaw dimensions, stature and sex in contemporary populations, it seems important to recognise that the associations observed reflect evolutionary trends. Over the course of evolution, changes in lifestyle and social behaviour have had far-reaching consequences for human dentition. The transition to a farming lifestyle in the Neolithic period, the effects of industrialisation on eating habits and the consumption of the highly processed diet that is favoured today represent the prominent factors in the reduction in tooth and jaw size dimensions. Western industrialised nations are most affected by this development. As a consequence of modern lifestyles, which minimise the functional importance of dentition for food intake, the maxillofacial region, including the soft-tissue structures in this area, and the teeth seem predestined to be morphologically and metrically remodelled even further in the future.

## Figures and Tables

**Table 1 biology-13-00569-t001:** The jaw and tooth dimensions analysed in this study (* Numeration acc. to Bräuer [34]; # for all teeth). Due to the high number of aplasia or extractions of third molars, the corresponding measurements were taken on second molars (traits no. 2., 7., 8.). Other modifications were made in relation to two dimensions (nos. 1., 6.).

No.	DIMENSION	NUMERATION *	MEASURING POINTS
1.	Internal palatal length	62	orale–staphylion (mod.: distal of the second molar)
2.	Palate end width	63 (1)	posterior endpoints of the palate in the third molar (mod.: second molar)
3.	Anterior palatal width	63 (2)	inner alveolar margin between canine and first premolar
4.	Palatal height	64	behind the first molars on the median sagittal plane
5.	Anterior palatal height	64a.	behind the first premolars on the median sagittal plane
6.	Anterior mandibular width	67	inner margin of both mental foramina (mod.: between the first and second premolars)
7.	Dental arch, length of maxilla	80	labial surface of the incisors from the centre of a straight line touching the distal surfaces of the (mod.) second molars
8.	Dental arch, length of mandible	80a	labial surface of the incisors from the centre of a straight line touching the distal surfaces of the (mod.) second molars
9.	External dental arch width	80 (1)	largest lateral extension of the rows of teeth perpendicular to the median sagittal plane (in the maxilla)
	Mesio-distal diameter #	81	maximum distance between the mesial and distal surfaces
	Bucco-lingual diameter #	81 (1)	maximum distance between the buccal and labial surfaces

**Table 2 biology-13-00569-t002:** Measurement errors of all tooth diameters (teeth 1–7 in the dental arch) and jaw dimensions (see Table 1). The standard errors (SEs) of the original single measurements and of the variables are shown. The latter is also expressed as a coefficient of variation (CV), i.e. as a percentage of the mean (MD = mesio-distal; BL = bucco-lingual; UJ = upper jaw; LJ = lower jaw; ICC = intra-class correlation coefficient). Tooth diameters in both the maxilla and the mandible were averaged over the left and right sides (see the Materials and Methods section).

Variable	SE of Single Measurements	SE of Variables	CV of Variables	ICC
MDUJ1	0.14	0.08	1.00	0.98
MDUJ2	0.17	0.11	1.68	0.96
MDUJ3	0.14	0.11	1.49	0.94
MDUJ4	0.14	0.09	1.36	0.95
MDUJ5	0.13	0.09	1.32	0.95
MDUJ6	0.15	0.11	1.05	0.95
MDUJ7	0.19	0.16	1.72	0.92
MDLJ1	0.11	0.07	1.28	0.96
MDLJ2	0.12	0.08	1.35	0.96
MDLJ3	0.15	0.09	1.48	0.95
MDLJ4	0.16	0.10	1.56	0.95
MDLJ5	0.13	0.11	1.55	0.94
MDLJ6	0.14	0.10	0.92	0.97
MDLJ7	0.18	0.13	1.25	0.96
BLUJ1	0.18	0.10	1.46	0.96
BLUJ2	0.18	0.17	2.77	0.88
BLUJ3	0.15	0.11	1.32	0.96
BLUJ4	0.09	0.07	0.73	0.98
BLUJ5	0.10	0.07	0.79	0.98
BLUJ6	0.12	0.08	0.71	0.98
BLUJ7	0.17	0.12	1.08	0.97
BLLJ1	0.18	0.09	1.60	0.96
BLLJ2	0.21	0.10	1.59	0.93
BLLJ3	0.14	0.10	1.39	0.96
BLLJ4	0.11	0.08	0.97	0.98
BLLJ5	0.12	0.10	1.21	0.96
BLLJ6	0.13	0.10	0.96	0.97
BLLJ7	0.13	0.10	1.02	0.97
1—Internal palatal length	0.44	0.31	0.78	0.98
2—Palate end width	0.15	0.11	0.27	1.00
3—Anterior palatal width	0.65	0.46	1.42	0.94
4—Palatal height	0.41	0.29	1.54	0.99
5—Anterior palatal height	0.40	0.28	1.82	0.98
6—Anterior mandibular width	0.52	0.37	0.83	0.97
7—Dental arch length of maxilla	0.48	0.34	0.65	0.98
8—Dental arch length of mandible	0.55	0.39	0.78	0.97
9—External dental arch width	0.28	0.20	0.33	1.00

**Table 3 biology-13-00569-t003:** Population distribution of tooth diameters (teeth 1–7 in the dental arch) and jaw dimensions by sex. For both sexes, the number of teeth (*n*) registered, the mean, and the standard deviation (sd) are shown. The ratio between the mean values in males/females is also shown (MD = mesio-distal; BL = bucco-lingual; UJ = upper jaw; LJ = lower jaw). Tooth diameters were averaged over the left and right sides.

	FEMALE			MALE			Ratio
Variables	*n*	Mean	sd	*n*	Mean	sd	
MDUJ1	50	8.1	0.49	50	8.5	0.53	1.04
MDUJ2	50	6.3	0.54	50	6.5	0.55	1.03
MDUJ3	50	7.1	0.36	50	7.5	0.43	1.06
MDUJ4	50	6.5	0.40	48	6.8	0.34	1.04
MDUJ5	49	6.4	0.38	50	6.6	0.38	1.04
MDUJ6	50	10.1	0.50	50	10.3	0.48	1.02
MDUJ7	50	9.2	0.54	50	9.6	0.55	1.05
MDLJ1	50	5.2	0.31	50	5.3	0.36	1.02
MDLJ2	50	5.6	0.31	50	5.8	0.40	1.04
MDLJ3	50	6.1	0.33	50	6.6	0.41	1.08
MDLJ4	50	6.4	0.42	49	6.7	0.41	1.05
MDLJ5	49	6.8	0.44	50	7.1	0.39	1.05
MDLJ6	50	10.4	0.52	50	10.7	0.58	1.03
MDLJ7	50	10.0	0.57	50	10.4	0.62	1.04
BLUJ1	50	6.9	0.53	50	7.2	0.51	1.04
BLUJ2	50	6.2	0.50	50	6.3	0.49	1.02
BLUJ3	50	7.9	0.48	50	8.4	0.47	1.07
BLUJ4	50	9.1	0.49	48	9.4	0.52	1.04
BLUJ5	49	9.2	0.58	50	9.5	0.52	1.03
BLUJ6	50	11.2	0.51	50	11.7	0.63	1.04
BLUJ7	50	11.0	0.60	50	11.6	0.67	1.05
BLLJ1	50	5.8	0.44	50	6.0	0.45	1.03
BLLJ2	50	6.1	0.36	50	6.2	0.41	1.01
BLLJ3	50	7.2	0.42	50	7.7	0.50	1.07
BLLJ4	50	7.6	0.43	49	7.9	0.55	1.04
BLLJ5	49	8.2	0.46	50	8.5	0.53	1.04
BLLJ6	50	10.2	0.51	50	10.6	0.51	1.03
BLLJ7	50	10.1	0.51	50	10.4	0.58	1.03
1—Internal palatal length	50	39.4	2.04	50	40.3	2.60	1.02
2—Palate end width	50	38.4	2.55	50	41.0	2.86	1.07
3—Anterior palatal width	50	31.5	1.66	50	32.9	1.80	1.05
4—Palatal height	50	17.6	2.07	50	20.4	2.65	1.16
5—Anterior palatal height	50	15.4	2.17	50	15.7	2.34	1.02
6—Anterior mandibular width	50	42.9	1.28	50	45.0	2.11	1.05
7—Dental arch length of maxilla	50	51.1	2.59	50	53.2	2.26	1.04
8—Dental arch length of mandible	50	49.1	1.95	50	50.8	2.09	1.03
9—External dental arch width	50	59.4	2.59	50	63.1	2.89	1.06

**Table 4 biology-13-00569-t004:** The correlation between stature and tooth diameters (teeth 1–7 in the dental arch) or jaw dimensions (see Table 1) (MD = mesio-distal; BL = bucco-lingual; UJ = upper jaw; LJ = lower jaw).

	MD		BL	
Tooth Number	UJ	LJ	UJ	LJ
1	0.42	0.31	0.36	0.38
2	0.26	0.38	0.30	0.30
3	0.49	0.52	0.47	0.54
4	0.42	0.41	0.42	0.41
5	0.39	0.40	0.36	0.44
6	0.25	0.32	0.45	0.35
7	0.42	0.32	0.46	0.30
1—Internal palatal length	0.34			
2—Palate end width	0.35			
3—Anterior palatal width	0.37			
4—Palatal height	0.47			
5—Anterior palatal height	0.04			
6—Anterior mandibular width	0.55			
7—Dental arch length of maxilla	0.43			
8—Dental arch length of mandible	0.40			
9—External dental arch width	0.51			

**Table 5 biology-13-00569-t005:** The variation in tooth crown diameters (teeth 1–7 in the dental arch) and jaw dimensions (see Table 1) in relation to stature, sex and both variables together. Also, the *p*-value for an additional impact of sex on top of stature is shown. R^2^ values above 0.15 are highlighted in grey and R^2^ values above 0.20 are highlighted in bold (MD = mesio-distal; BL = bucco-lingual; R^2^ = determination coefficient).

	Upper Jaw				Lower Jaw			
Tooth Diameter/Jaw Dimension	R^2^ Stature	R^2^ Sex	R^2^ Stature + Sex	*p*-Value Sex	R^2^ Stature	R^2^ Sex	R^2^ Stature + Sex	*p*-Value Sex
MD1	0.17	0.09	0.16	0.965	0.09	0.02	0.09	0.288
MD2	0.06	0.01	0.05	0.429	0.14	0.07	0.13	0.917
MD3	**0.23**	**0.20**	**0.24**	0.160	**0.26**	**0.29**	**0.31**	0.007
MD4	0.17	0.09	0.16	0.915	0.16	0.12	0.15	0.452
MD5	0.14	0.10	0.13	0.564	0.15	0.14	0.16	0.211
MD6	0.05	0.03	0.04	0.874	0.10	0.06	0.09	0.708
MD7	0.17	0.12	0.16	0.431	0.09	0.11	0.11	0.117
BL1	0.12	0.07	0.11	0.841	0.14	0.03	0.15	0.139
BL2	0.08	0.01	0.09	0.207	0.08	0.00	0.13	0.012
BL3	**0.21**	**0.23**	**0.24**	0.027	**0.28**	**0.20**	**0.28**	0.330
BL4	0.17	0.10	0.16	0.905	0.16	0.10	0.15	0.809
BL5	0.12	0.06	0.12	0.764	0.19	0.10	0.18	0.979
BL6	0.19	0.14	0.19	0.482	0.11	0.10	0.11	0.275
BL7	**0.20**	0.16	**0.20**	0.305	0.08	0.05	0.07	0.650
1—Internal palatal length	0.11	0.02	0.12	0.182				
2—Palate end width	0.11	0.19	0.18	0.003				
3—Anterior palatal width	0.13	0.14	0.14	0.083				
4—Palatal height	**0.21**	**0.25**	**0.26**	0.008				
5—Anterior palatal height	−0.01	−0.01	−0.02	0.713				
6—Anterior mandibular width	**0.30**	**0.25**	**0.31**	0.089				
7—Dental arch length of maxilla	0.18	0.14	0.18	0.344				
8—Dental arch length of mandible	0.15	0.14	0.16	0.133				
9—External dental arch width	**0.26**	**0.31**	**0.32**	0.002				

**Table 6 biology-13-00569-t006:** Correlation between tooth size diameters (teeth 1–7 in the dental arch) and jaw dimensions (see Table 1). Correlations above 0.5 are marked in dark grey and highlighted in bold. Correlations above 0.3 are marked in light grey (MD = mesio-distal; BL = bucco-lingual; UJ = upper jaw; LJ = lower jaw).

Tooth	1	2	3	4	5	6	7	1	2	3	4	5	6	7
	BL—UJ							BL—LJ						
1—Internal palatal length	0.27	0.44	0.39	**0.51**	**0.51**	0.42	0.40	0.34	0.30	0.41	0.46	0.44	0.25	0.47
2—Palate end width	0.01	0.04	−0.12	0.02	−0.03	0.05	−0.07	−0.05	−0.06	−0.02	−0.05	−0.09	−0.03	−0.05
3—Anterior palatal width	0.01	0.14	−0.11	−0.04	−0.17	0.04	−0.03	0.01	0.11	0.02	−0.03	−0.19	0.07	−0.06
4—Palatal height	0.11	0.19	0.10	0.11	0.16	0.06	0.05	0.17	0.10	0.14	−0.07	0.03	0.06	0.07
5—Anterior palatal height	0.02	0.15	0.00	0.03	0.03	−0.11	−0.12	0.07	−0.11	−0.02	−0.11	0.02	0.04	−0.06
6—Anterior mandibular width	0.19	0.32	0.24	0.40	0.41	0.46	0.36	0.28	0.27	0.28	0.43	0.37	0.42	0.30
7—Dental arch length of maxilla	0.38	**0.52**	0.44	**0.55**	**0.57**	**0.51**	0.44	0.36	0.36	**0.50**	0.49	0.42	0.35	0.48
8—Dental arch length of mandible	0.39	0.49	0.42	**0.60**	**0.62**	**0.58**	**0.57**	0.48	0.42	**0.51**	**0.58**	**0.59**	0.38	**0.52**
9—External dental arch width	0.17	0.23	0.10	0.24	0.25	0.39	0.36	0.17	0.17	0.24	0.16	0.15	0.24	0.25
1st latent factor score	0.04	0.07	−0.08	0.05	0.01	0.17	0.09	−0.00	0.02	0.04	−0.00	−0.05	0.07	0.03
2nd latent factor score	0.39	**0.54**	0.49	**0.62**	**0.64**	**0.57**	**0.56**	0.44	0.41	**0.55**	**0.57**	**0.55**	0.40	**0.56**
	MD—UJ							MD—LJ						
1—Internal palatal length	0.39	0.49	**0.58**	**0.63**	0.41	**0.56**	**0.59**	0.34	0.36	0.45	**0.50**	**0.50**	0.33	**0.57**
2—Palate end width	0.21	0.11	0.00	0.08	0.04	−0.02	−0.08	0.10	0.04	0.09	0.14	0.01	0.00	−0.01
3—Anterior palatal width	0.18	0.13	0.20	0.00	−0.12	0.12	0.13	0.17	0.19	0.25	0.02	−0.09	0.07	0.01
4—Palatal height	0.27	0.15	0.00	0.21	0.15	−0.14	0.03	0.16	0.11	0.08	0.22	0.09	0.02	0.06
5—Anterior palatal height	0.19	0.13	−0.07	0.26	0.19	−0.03	0.10	0.12	0.02	−0.01	0.21	0.09	−0.03	0.05
6—Anterior mandibular width	0.35	0.39	0.44	0.37	0.38	0.30	0.31	0.32	0.34	0.42	0.33	0.38	0.36	0.40
7—Dental arch length of maxilla	0.47	**0.53**	**0.68**	**0.62**	**0.50**	**0.58**	**0.53**	0.37	0.37	**0.54**	**0.54**	**0.55**	0.41	**0.56**
8—Dental arch length of mandible	0.44	0.41	**0.60**	**0.57**	**0.55**	**0.59**	**0.64**	0.47	0.47	**0.57**	**0.53**	**0.66**	**0.52**	**0.70**
9—External dental arch width	0.33	0.24	0.22	0.24	0.21	0.26	0.20	0.24	0.20	0.28	0.22	0.23	0.24	0.28
1st latent factor score	0.21	0.12	0.05	0.06	0.04	0.07	−0.01	0.12	0.08	0.14	0.09	0.03	0.08	0.05
2nd latent factor score	0.46	**0.53**	**0.70**	**0.66**	**0.53**	**0.65**	**0.66**	0.42	0.43	**0.57**	**0.56**	**0.62**	0.47	**0.68**

## Data Availability

The anonymised original data were curated by the study director only. A data file (.xls) with the original raw data will be available on request from the corresponding author after the publication of this paper.

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
