# Peer review of "The Correlation of Tooth Sizes and Jaw Dimensions with Biological Sex and Stature in a Contemporary Central European Population"

_biology, 2024, doi:10.3390/biology13080569_

Round 1
Reviewer 1 Report
Comments and Suggestions for Authors
This manuscript summarizes research on the relationship of various dental measurements with biological sex and stature. A few individual tooth measurements and jaw dimension measurements demonstrated a correlation with sex and stature, with a greater correlation with stature when considering raw data. Although it can be extrapolated that the correlation with stature is a function of biological sex since stature is sexually dimorphic, more research is needed on a wider swath of the population than the dataset used in this research. The research highlights the high evolutionary component to dental traits demonstrated throughout the prevailing literature and suggests additional lines of research in the presentation of tooth and jaw measures seen in current populations.
Overall, the manuscript presents solid results of an interesting study of dental remains measurements. The research provides additional evidence of sexual dimorphism within tooth and jaw measurements using a contemporary European population. The correlation analysis at the core of the research suggests linkages between measures, sex, and stature, but more research using multivariate analyses is needed to evaluate causality.
There are no concerns with the analytical methods applied or the results obtained. There are several areas that need to be reviewed during copyediting for clarity of the English, which are not detailed below. Below outlines areas of concern and/or areas to address to improve the manuscript.
Lines 38-40- Premise of the manuscript is highlighting the correlation of tooth size and jaw dimensions with sex and stature. In the sentence “Our results…” there is a suggestion of causality vice correlation. Suggest rephrasing this sentence to a hypothesize instead of fact. The data suggests this but does not prove this.
Line 49- word “already: unnecessary
Lines 103-108: Table 1. This table provides an overview of the materials used, so would be better situated within section 2 under materials and method.
Lines 103-108: Table 1: description of measure points for No. 7 and 8, suggest adding “3rd molar” to definition to be consistent with what is stated in No. 2. This makes it clearer on what was changed within the table.
Line 111- citations- I believe the citations here should be 23-27 and the Line 113 citations should be 28-30. The reference 27 focus is on sexual dimorphism and not stature.
Line 114- missing “the”. Should state “provide the most”
Lines 109-117- consider moving this paragraph before previous paragraph at Line 90. This would improve flow of introduction. It seems out of place in its current location.
Line 126- recommend adding (n=2 and n=3) next to Russia and Iran to make it clear you are referring to sample size and not a reference.
Lines 201-203- incomplete sentence
Line 239-241- Revise sentence for clarity. “did not differ” is unclear what you are referring to. Are you referring to “significant difference” as demonstrated by a statistical test or are you talking about raw difference? If raw difference you cannot state they did not differ.
Lines 303-304- word missing at the end of this sentence. Believe this should be “…environmental factors.”
Lines 374-375- The argument is unclear. Please revise since I do not know what you are trying to say.
Lines 391-392- Language needs to be cleaned up for clarity.
Lines 397-398- Suggest revising sentence to start as “This suggests the first factor…”
Line 455- when saying “their highly quantifiable traits” Are you talking about their nature or characters? I would advise not to use traits here.
Lines 503-505- You have not made a strong enough argument to support such a definitive statement. With more research specifically evaluating and testing this hypothesis but the limited sample of this study is not enough to recommend usage for stature determinations.
Section 3.9: Suggesting making the strengths and weakness point more clear as to what were the strengths and what were the weaknesses. This section as written sounds more like discussion and conclusions as written.
Comments on the Quality of English LanguageRecommend proof-reading manuscript for English language. Most concerns are copy-editing issues. There are several sentences that need punctuation to improve the clarity of what you are trying to say. For example. Line 224 should have a semicolon after “canines.” Also, several sentences have extraneous words that detract from what is being said. For example, line 220 “which could be confirmed in our study” could simply be stated as “which our study confirmed”
Author Response
Reviewer 1
Point-by-point response
We would like to thank you for the critical review that we have implemented in all points. The entire manuscript was finally completely revised by a native speaker.
Lines 38-40- Premise of the manuscript is highlighting the correlation of tooth size and jaw dimensions with sex and stature. In the sentence “Our results…” there is a suggestion of causality vice correlation. Suggest rephrasing this sentence to a hypothesize instead of fact. The data suggests this but does not prove this.
We revised this in the following way:
Our results suggest that the differences between the sexes in most crown diameters and some jaw dimensions may be related to the stature of the individuals.
Line 49- word “already: unnecessary
The word has been deleted
Lines 103-108: Table 1. This table provides an overview of the materials used, so would be better situated within section 2 under materials and method.
Table 1 has been moved to chapter Material and Methods
Lines 103-108: Table 1: description of measure points for No. 7 and 8, suggest adding “3rd molar” to definition to be consistent with what is stated in No. 2. This makes it clearer on what was changed within the table.
The description of the measuring points in Table 1 has been corrected
Line 111- citations- I believe the citations here should be 23-27 and the Line 113 citations should be 28-30. The reference 27 focus is on sexual dimorphism and not stature.
Thank you very much for the correction. We have changed the references accordingly.
Line 114- missing “the”. Should state “provide the most”
We have changed this accordingly (now line 107)
Lines 109-117- consider moving this paragraph before previous paragraph at Line 90. This would improve flow of introduction. It seems out of place in its current location.
Thanks for the suggestion.
The change has become superfluous due to the conversion of Table 1
Line 126- recommend adding (n=2 and n=3) next to Russia and Iran to make it clear you are referring to sample size and not a reference.
The changes were made accordingly (now line 119)
Lines 201-203- incomplete sentence
Thanks for the important proposal. We revised this in the following way: (now line 208)
The sentence has been completed. …… the countries from which the majority of students hailed [37].
Line 239-241- Revise sentence for clarity. “did not differ” is unclear what you are referring to. Are you referring to “significant difference” as demonstrated by a statistical test or are you talking about raw difference? If raw difference you cannot state they did not differ.
The sentence has been amended as follows: (now line 243)
The association between stature and crown diameter is very similar between the upper and lower jaws (average correlations of 0.39 and 0.38, respectively), or between BL and MD diameters (average correlations of 0.40 and 0.38, respectively).
Lines 303-304- word missing at the end of this sentence. Believe this should be “…environmental factors.”
The missing word was added to the sentence. (now line 308)
Lines 374-375- The argument is unclear. Please revise since I do not know what you are trying to say.
The paragraph has been reworded as follows: (now lines 376-379)
The jaw dimensions internal palatal length, anterior mandibular width, and dental arch length loading on the second factor are all related to the space available for the teeth. The fact that the second factor is distinctly correlated with the tooth diameters suggests that the jaw acts as a place-holder for the teeth.
Lines 391-392- Language needs to be cleaned up for clarity.
The sentences have been revised as follows: (now lines 392-395)
The two latent factors also vary with regard to the degree of sexual dimorphism. The four jaw dimensions loading on the second factors show ratios between 1.02 and 1.05 (Table 3), i.e., a limited degree of sexual dimorphism compared to other jaw dimensions. In addition, the data failed to demonstrate an impact of sex on these dimensions beyond the impact of stature (Table 5).
Lines 397-398- Suggest revising sentence to start as “This suggests the first factor…”
The sentence has been rephrased: (now lines 400-401)
This suggests that the first factor must be interpreted as perceived facial masculinity [69], which does, in fact, interact with stature [70].
Line 455- when saying “their highly quantifiable traits” Are you talking about their nature or characters? I would advise not to use traits here.
The sentence has been rephrased: (now lines 358-360)
Their features are easy to quantify and the fact that teeth are formed over a short period of time means that environmental factors have a limited impact.
Lines 503-505- You have not made a strong enough argument to support such a definitive statement. With more research specifically evaluating and testing this hypothesis but the limited sample of this study is not enough to recommend usage for stature determinations.
We agree with the reviewer that this is a question requiring further research. We rephrased this in the following way. (now lines 500ff)
Our study demonstrates that a systematic analysis of the association between tooth size, jaw dimensions, stature and biological sex can provide a series of insights: (1) Tooth crown diameters and jaw dimensions show a pronounced sexual dimorphism; (2) Crown diameters and jaw dimensions correlate with both stature and biological sex; (3) Stature seems to be the main mediating factor in the influence of biological sex on dental metrics; (4) The sexual dimorphism of teeth remains a decisive factor in the differences observed. The results of the jaw dimensions suggest that they may be as useful, or perhaps even more useful, than crown diameters when it comes to sex and stature determinations. Further research will be required to clarify their value both in this and in other respects.
Section 3.9: Suggesting making the strengths and weakness point more clear as to what were the strengths and what were the weaknesses. This section as written sounds more like discussion and conclusions as written.
Section 3.9 and the 4. Conclusion have been completely revised to better separate the two chapters in terms of content.
Conclusions:
Accurate sex and stature determinations play a fundamental role in biological and medical applications and research. Our study has improved the approach to formal reconstructions and the ability to interpret individual estimates. When interpreting associations between tooth sizes, jaw dimensions, stature and sex in contemporary populations, it seems important to recognise that the associations observed reflect evolutionary trends. Over the course of evolution, changes in lifestyle and social behaviour had far-reaching consequences for human dentition. The transition to a farming lifestyle in the Neolithic period, the effects of industrialisation on eating habits and the consumption of the highly processed diet that is favoured today represent the prominent factors in the reduction of tooth and jaw size dimensions. The western industrialised nations are most affected by this development. As a consequence of modern lifestyles, which minimise the functional importance of dentition for food intake, the maxillofacial region, including the soft-tissue structures involved, and the teeth seem predestined to be morphologically and metrically remodelled even further in the future.
Comments on the quality of English language
The entire manuscript and all table captures has been revised by a native speaker

Reviewer 2 Report
Comments and Suggestions for Authors
This manuscript is an excellent study on dental and gnathic variation in a modern population. I look forward to seeing it in print. I really enjoy reading about modern skeletal biology and the exploration of character correlation. I think this contribution is really important in providing modern data on a younger population. The trends these authors have documented will be interesting to further explore with larger datasets. This paper does provide an excellent foundation for more expansive studies. I have only a few suggestions and they are only text edits. I am noting them by line number:
105-106: In the figure caption there are periods after 1 and 6 but not 2, 7, 8.
530: do you mean cues or clues? Cues sounded odd as I read it, but it does work in the context of the sentence.
791: Table caption: You have written Partial led. Do you mean partialed?
I enjoyed this paper a great deal it was very clearly and concisely written. I hope to see it in print soon.
Author Response
Reviewer 2
Point-by-point response
We would like to thank you for the critical review that we have implemented in all points. The entire manuscript was finally completely revised by a native speaker.
Lines 105-106: In the figure caption there are periods after 1 and 6 but not 2, 7, 8
We have changed this in the figure caption Table 1
Line 530: do you mean cues or clues? Cues sounded odd as I read it, but it does work in the context of the sentence.
Thank you for pointing out this typo.
The conclusion has been revised and the sentence has been deleted.
Line 791: Table caption: You have written Partial led. Do you mean partialed?
Thank you for the important tip. We have corrected this spelling mistake. (now line 787)